# NEURO-CAUSAL FACTOR ANALYSIS

## ABSTRACT

Factor analysis (FA) is a statistical tool for studying how observed variables with some mutual dependences can be expressed as functions of mutually independent unobserved factors, and it is widely applied throughout the psychological, biological, and physical sciences. We revisit this classic method from the comparatively new perspective given by advancements in causal discovery and deep learning, introducing a framework for *Neuro-Causal Factor Analysis (NCFA)*. Our approach is fully nonparametric: it identifies factors via latent causal discovery methods and then uses a variational autoencoder (VAE) that is constrained to abide by the Markov factorization of the distribution with respect to the learned graph. We evaluate NCFA on real and synthetic data sets, finding that it performs comparably to standard VAEs on data reconstruction tasks but with the advantages of sparser architecture, lower model complexity, and causal interpretability. Unlike traditional FA methods, our proposed NCFA method allows learning and reasoning about the latent factors underlying observed data from a justifiably causal perspective, even when the relations between factors and measurements are highly nonlinear.

## 1 INTRODUCTION

Since its development over a century ago, factor analysis (FA) (Spearman, 1904) [1] has been applied in many scientific fields, including genomics, computational biology (Pournara & Wernisch, 2007; Velten et al., 2022), economics (Forni & Reichlin, 1998; Ludvigson & Ng, 2007), sociology (Bollen, 2012) and many others. The goal of FA is to offer explanations of variability among dependent observables via (potentially) fewer latent variables that capture the degree to which the observables in the system vary jointly. For the sake of identifiability, it is common to assume linearity, although in practice it is well-known that many problems exhibit complex nonlinear latent structures. With the rise of nonparametric deep generative models that allow representing highly nonlinear relationships between dependent observables, one might hope to combine the best of both worlds.

Moreover, within applications such as those listed above, FA is considered useful because the learned factors (latents) may offer a possible interpretation of relevant observed correlations. Many applied FA studies provide an interpretation of the learned factors based on the observed variables whose joint correlation they encode. A natural tendency when trying to interpret these factors is to assume they reflect possible common causes linking observed variables. However, the models used in such studies are not necessarily built with causality in mind. Collectively, these considerations purport a need for a framework for nonlinear *causal* factor analysis that combines identifiability with flexibility through the use of modern advances in deep generative models and causality.

To this end, we propose *Neuro-Causal Factor Analysis (NCFA)*, augmenting classic FA on both fronts by leveraging advancements of the last few decades, including (i) causal discovery (Spirtes et al., 2000; Pearl, 2009) and (ii) deep generative models, such as variational autoencoders (VAEs) (Kingma & Welling, 2014). To formalize this combination of ideas and apply it to the settings where FA is typically invoked, we consider causal models that directly abide by *Reichenbach's common cause principle* (Reichenbach, 1956, p. 157): dependent variables in a system that do not share a direct causal relation should be explained by the existence of one or more unobserved common causes which when conditioned upon render them independent. In particular, NCFA is applicable to

---

[1] We would like to briefly draw the reader's attention to and repudiate the historical context within which factor analysis and related methods were originally developed (e.g. Saini, 2019; Crenshaw et al., 1995; Stubblefield, 2007).

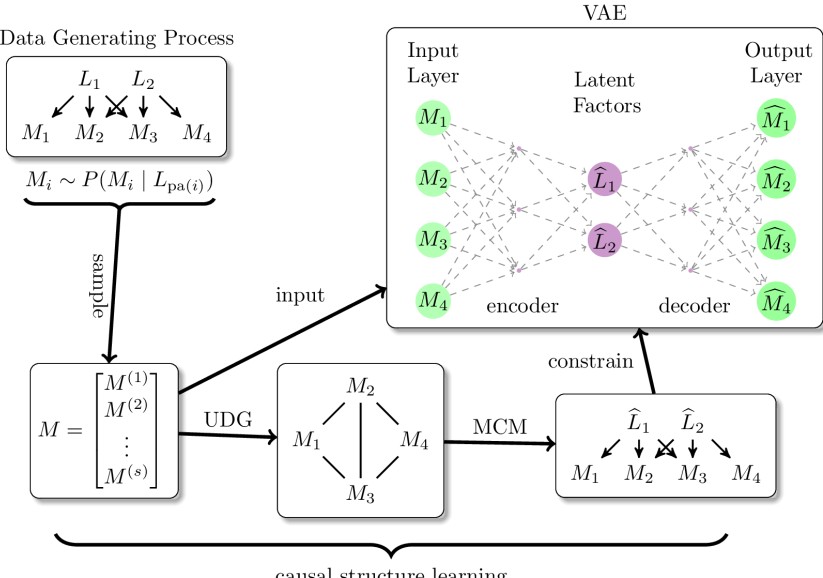

Figure 1: Pipeline for learning a neuro-causal factor model. Given a sample from a suitable data generating process, NCFA estimates a causal structure, which it then uses to constrain a VAE that it trains on the sample, resulting in a causally-interpretable deep generative model.

problems where one can assume that the observed (or *measurement*) variables are rendered mutually independent when conditioning on a set of unobserved latent variables, which may be interpreted as *causally justifiable* factors from the FA perspective. Such models naturally arise, for instance, when one wishes to interpret causal relations among pixel variables in image data, such as biomedical imaging data. In these contexts, each pixel in the image is treated as a random variable that may be dependent with other pixels. Since pixels should have no direct causal relations, all dependences should be explained by the latent information (for instance, neuronal activity in the brain during an fMRI scan) which resulted in the observed pixel intensities. In such situations, the common cause principle naturally applies.

Our main contribution is the NCFA framework (Figure 1) for causally interpretable, identifiable FA models with the flexibility and data replication capabilities afforded by deep generative models. Our approach does not assume the underlying structure is known (i.e. it is learned from data), allows for flexible estimation of the latent space with deep generative models, and comes with fully nonparametric (i.e. no functional assumptions are imposed) identifiability guarantees. One of the key methodological contributions is the introduction of *latent degrees of freedom* whereby additional representational capacity is afforded by giving each causal variable its own factorial prior. We demonstrate on both synthetic and real data that NCFA injects generative models with interpretable structure without any significant loss of representational or predictive capacity compared to unstructured generative models. Moreover, we provide an algorithm and open source implementation for inference and prediction with NCFA models.

The paper is organized as follows: We begin in Section 2 with a survey of related work in factor analysis, latent causal modeling, and deep generative models. In Section 3, we formally define NCFA models and present identifiability results. Next, in Section 4, we provide the NCFA algorithm and discuss its complexity. We then conclude by comparing NCFA to ground truth causal models and baseline VAE methods on synthetic and real data sets in Section 5.

## 2 COMPARISON TO RELATED WORK

We divide the vast amount of related work into three areas: factor analysis (Section 2.1), latent causal discovery (Section 2.2), and deep generative models (Section 2.3). Before describing each in more

detail in its respective subsection, we first summarize their differing motivations and methods and provide a comparison to our proposed NCFA.

Factor analysis focuses on modeling measurement variables in terms of underlying factors (which can be interpreted as sources), focusing on model simplicity and interpretability, generally by assuming linear relations and jointly Gaussian random variables. Latent causal models focus on more detailed causal structure, not being limited to measurement variables and their latent sources, resulting in extremely interpretable models, but often at the expense of (arguably) strong, untestable assumptions like faithfulness. Deep generative models focus on learning as accurate a black box model as possible, optimizing a highly overparamaterized and nonlinear model to still achieve generalizability. Although the interpretation of deep generative models as nonlinear factor analysis is standard in the literature (e.g. Roweis & Ghahramani, 1999; Murphy, 2022; Goodfellow et al., 2016), the additional dimensions of causality and identifiability are new to our approach. NCFA offers a unifying perspective on structured representation learning, incorporating the strengths of each of these approaches.

Like FA, we focus on modeling measurement variables in terms of their underlying sources; however, NCFA identifies these sources and their structural connections to the measurements through explicit latent causal structure learning, which is made easier and requires weaker assumptions by focusing on source-measurement causal relations instead of more detailed intermediate causal structure. Furthermore, the source distributions and their corresponding functional relations to the measurements are estimated using a VAE whose architecture is constrained to respect the learned causal structure, gaining some of the expressiveness of deep generative models but regularized to maintain causal interpretability and generalizability. Hence, NCFA is motivated by the simplicity of FA, the causal interpretability of latent causal models, and the expressive power of deep generative models.

## 2.1 FACTOR ANALYSIS

We now give a brief introduction to FA, focusing on the key terms and mathematical ideas [2] that we connect to latent causal discovery and deep generative models, but for a more in-depth introduction and discussion about FA, see Mulaik (2009).

**Definition 2.1.** A *factor model* represents a random (row) vector $\mathbf{M} \sim \mathcal{N}(\mathbf{0}, \mathbf{\Sigma})$ consisting of $n$ *measurement variables* as a linear transformation of a standard jointly normal random vector $\mathbf{L} \sim \mathcal{N}(\mathbf{0}, I_K)$ of $K < n$ *latent factors* via *factor loading weights* $W \in \mathbb{R}^{K \times n}$ plus a jointly normal random vector of $n$ *error terms* $\boldsymbol{\epsilon} \sim \mathcal{N}(\mathbf{0}, D)$, where $D \in \mathbb{R}_+^{n \times n}$ is a diagonal matrix, via

$$\mathbf{M} = \mathbf{L}W + \boldsymbol{\epsilon}.$$

Given a sample $M \in \mathbb{R}^{s,n} \sim \mathbf{M}$ and assuming that $\mathbf{L}$ and $\boldsymbol{\epsilon}$ are probabilistically independent, the factor model can be estimated (Adachi, 2019) from the empirical covariance matrix $\widehat{\Sigma} = \frac{1}{s} M^\top M$ by finding $\widehat{W}$ and $\widehat{D}$ that minimize the squared Frobenius norm

$$\|\widehat{\Sigma} - \widehat{W}^\top \widehat{W} - \widehat{D}\|_F^2.$$

Such a solution is unique only up to orthogonal transformations of $\widehat{W}$, and so without further (e.g., in our case, causal) assumptions, finding a solution does not always warrant a meaningful interpretation of the resulting factor model. This unidentifiability poses a problem in *exploratory* FA, where there is no prior knowledge about $\widehat{\Sigma}, \widehat{W}$ or $\widehat{D}$, but less so in *confirmatory* FA, where experts incorporate domain knowledge to constrain and interpret solutions as well as test specific hypotheses.

Additionally, there are possibilities for either restricting or relaxing the FA model, including closely related methods like PCA (Pearson, 1901; Hotelling, 1933; Jolliffe, 2002), ICA (Comon, 1994; Hyvärinen & Oja, 2000), and many others beyond our scope. Notably, compared to other related work, sparse FA (Ning & Georgiou, 2011; Trendafilov et al., 2017; Yamamoto et al., 2017), which penalizes $\widehat{W}$ according to the number of nonzero entries, produces solutions more closely related to those we find with NCFA. The two main differences between sparse FA and NCFA are that (i) rather than explicitly penalizing the solution to encourage sparsity, NCFA simply learns a causal structure that exhibits a structure typically sought in sparse FA, and (ii) like most FA methods, sparse FA still assumes linearity and Gaussianity, whereas NCFA can be highly nonlinear and nonparametric.

---

[2]In case of conflicting notational conventions, e.g., $L$ to denote a loading matrix in FA literature versus denoting a set of latent variables in the causal graphical model literature, we favor the latter.

## 2.2 LATENT CAUSAL MODELS

Graphical causal modeling (Spirtes et al., 2000; Pearl, 2009) focuses on learning a directed acyclic graph (DAG) representation of the causal relations among variables. This typically requires a strengthening of the common cause principle (into what is sometimes called the causal Markov assumption), which additionally assumes causal sufficiency, i.e., that there are no latent variables, and hence that all probabilistic dependences among the observed variables are due to causal relations among them. Methods for learning latent causal models have classically focused on learning DAG-like structure (using mixed instead of only directed graphs) among the observed variables to the extent allowed by confounding latent variables, exemplified by algorithms such as FCI (Spirtes et al., 2000; Colombo et al., 2012) and IC (Pearl & Verma, 1995), which relax the Causal Markov Assumption. We also mention early work on this problem by (Martin & VanLehn, 1995; Friedman et al., 1997; Elidan et al., 2000). In contrast, research on causal measurement models (Silva et al., 2003) is more closely related to the goal of FA, in that it too focuses on factor-measurement relations. Recently, there has been a surge of interest in these models, with advances leveraging additive noise models (Maeda & Shimizu, 2021; Yang et al., 2022; Huang et al., 2022; Xie et al., 2022; Ashman et al., 2022), independent mechanisms (Gresele et al., 2021), weak supervision (Liu et al., 2022; Brehmer et al., 2022), and interventions (Chalupka et al., 2015; 2017; Ahuja et al., 2022; Squires et al., 2023; Varici et al., 2023).

## 2.3 STRUCTURED DEEP GENERATIVE MODELS

The past decade has seen a flurry of work on training large-scale deep latent variable models, fueled by advances in variational inference and deep learning (e.g. Larochelle & Murray, 2011; Kingma & Welling, 2014; Rezende et al., 2014; Dinh et al., 2014; Goodfellow et al., 2014; Rezende & Mohamed, 2015; Sohl-Dickstein et al., 2015). More recently, there has been a trend towards *structured* latent spaces, such as hierarchical, graphical, causal, and disentangled structures. Conceptually, NCFA provides a theoretically principled approach to automatically *learning* latent structure from data in a way that is causally meaningful. The related work here needs to be divided into two categories: *known* (e.g. from prior knowledge) vs. *learned* latent structure. These can be further divided into *non-causal* vs. *causal* approaches. Given that our main contribution is *learned causal structure*, we will focus the discussion on the latter: For causal structure, *identifiability* becomes crucial, as it is well-known that nonparametric latent variable models are unidentifiable in general (Hyvärinen & Pajunen, 1999; Locatello et al., 2019).

**Known structure**  Early work looked at incorporating known structure into generative models, such as autoregressive, graphical, and hierarchical structure (Germain et al., 2015; Johnson et al., 2016; Sønderby et al., 2016; Webb et al., 2018; Weilbach et al., 2020; Ding et al., 2021; Mouton & Kroon, 2023). This was later translated into known *causal* structure (Kocaoglu et al., 2017).

**Learned structure**  When the latent structure is unknown, several techniques have been developed to automatically learn useful (not necessarily causal) structure from data (Li et al., 2019; He et al., 2019; Wehenkel & Louppe, 2021; Kivva et al., 2022; Moran et al., 2023). More recently, based on growing interest in disentangled (Bengio, 2013) and/or causal (Schölkopf et al., 2021) representation learning, methods that automatically learn causal structure have been developed (Moraffah et al., 2020; Yang et al., 2021; Ashman et al., 2022; Shen et al., 2022; Kaltenpoth & Vreeken, 2023). Subramanian et al. (2022) assumes a linear Gaussian additive noise model, whereas Moraffah et al. (2020) uses GANs. Unlike NCFA, neither Moraffah et al. (2020) nor Subramanian et al. (2022) come with identifiability guarantees. In order to guarantee identifiability, CausalVAE (Yang et al., 2021) leverages additional labeled data $u$, based on iVAE (Khemakhem et al., 2020). DEAR (Shen et al., 2022) requires a known causal ordering, leaving "causal discovery from scratch to future work". More recently, Ashman et al. (2022) used partially additive models and Kaltenpoth & Vreeken (2023) used post-nonlinear models to guarantee identifiability. In contrast to this existing work, NCFA admits nonparametric identifiability guarantees without additional labels, known causal ordering, or specifying a particular parametric or functional form (see subsection 3.2).

## 3 NCFA MODELS

Consider a collection of jointly distributed measurement variables $(M_1, \ldots, M_n)$ for which we assume that all dependences are explained by the existence of a latent common cause of the measured

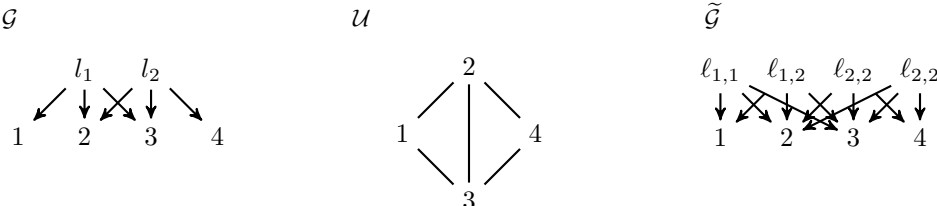

Figure 2: A UDG $\mathcal{U}$ associated with measurement variables $M_1, M_2, M_3, M_4$ and a corresponding minimum MCM graph $\mathcal{G}$ for the minimum edge clique cover $\mathcal{C} = \{C_1 = \{1, 2, 3\}, C_2 = \{2, 3, 4\}\}$. $\widetilde{\mathcal{G}}$ is the NCFA-graph for $\mathcal{G}$ with $\lambda = 4$ latent degrees of freedom. Note that all three graphs encode the exact same set of (marginal) independencies among the measurement variables $M_1, M_2, M_3, M_4$.

variables, i.e., that no $M_i$ and $M_j$ share a direct casual relation. If we were able to observe these latent confounders and condition upon them, $M_1, \ldots, M_n$ would become mutually independent. Hence, the only causal structure encoded via conditional independence in the observed distribution is contained in their marginal independence structure, which can be encoded in an undirected graph:

**Definition 3.1.** The *unconditional dependence graph* (UDG) for the jointly distributed random variables $(M_1, \ldots, M_n)$ is the undirected graph $\mathcal{U}$ with node set $[n] = \{1, \ldots, n\}$ and edge set

$$E = \{i - j : M_i \not\perp\!\!\!\perp M_j\}.$$

To recover a causal interpretation of the relations that hold among the measurement variables, we extend a UDG graph to a *(minimum) MCM graph*. Following the principle of Occam's Razor, we would like to explain the observed dependences in $(M_1, \ldots, M_n)$ in the simplest possible way, i.e., using the fewest possible latents to serve as the common causes of the measurement variables that exhibit dependence. To do so, we identify a *minimum edge clique cover* of the UDG $\mathcal{U}$, which is a collection $\mathcal{C} = \{C_1, \ldots, C_K\}$ of *cliques* (i.e., complete subgraphs of $\mathcal{U}$) such that for every $i - j \in E$ the pair $i, j$ is contained in at least one clique in $\mathcal{C}$ and there exists no set of cliques with this property that has cardinality smaller than $|\mathcal{C}|$.

**Definition 3.2.** Let $\mathcal{U}$ be an undirected graph with minimum edge clique cover $\mathcal{C} = \{C_1, \ldots, C_K\}$. The *(minimum) MCM graph* $\mathcal{G}$ for $\mathcal{U}$ and $\mathcal{C}$ is the DAG with vertices $[n] \cup L$ where $L = \{l_1, \ldots, l_K\}$ and edge set

$$E = \{l_i \to j : j \in C_i, \forall i \in [K]\}.$$

We call $|L|$ the number of *causal degrees of freedom* of the model.

An example of a UDG and a corresponding MCM graph is presented in Figure 2. Minimum MCM graphs were originally defined in the context of *MeDIL causal models* (Markham & Grosse-Wentrup, 2020). A summary of this theory is given in Appendix A, for completeness.

Since we assumed all marginal dependencies in $(M_1, \ldots, M_n)$ are explainable by the existence of a latent common cause, then the observed distribution $(M_1, \ldots, M_n)$ is realizable as the marginal distribution of $(M_1, \ldots, M_n)$ in the joint distribution $(M_1, \ldots, M_n, L_1, \ldots, L_K)$ that is Markov to the DAG $\mathcal{G}$, where $L_i$ is the random variable represented by the node $l_i$ in $\mathcal{G}$. From a factor analysis perspective, the latents $L_1, \ldots, L_K$ are the factors to be inferred.

### 3.1 NCFA GRAPHS AND VARIATIONAL AUTOENCODERS

The minimum MCM graph defines a putative causal graph that respects the independence structure of $(M_1, \ldots, M_n)$, and our goal is to learn the associated latent representations from data using a deep generative model. Consider the DAG $\mathcal{G}$ depicted in Figure 2 with two latents. A naïve approach would be to design a standard VAE such that the decoder respects the Markov properties implied by $\mathcal{G}$, however, it is unlikely that any generative model trained with a two-dimensional latent space will be able to represent the measurement variables accurately. The difficulty is that although the *true* causal structure involves only two latent variables, exactly fitting such a model is very difficult in practice. Thus, there is a tension between expressive capacity and respecting the causal structure.

We overcome this difficulty by replacing each causal latent with an overparametrized, factorial prior. The virtue of overparametrization is well-documented in the literature (Radhakrishnan et al.,

2020; Buhai et al., 2020); in our setting this has the effect of increasing representational capacity without breaking the Markov structure encoded in $\mathcal{G}$. Formally, given a minimum MCM graph $\mathcal{G} = \langle [n] \cup L, E \rangle$, we replace each $l_i$ with a set of independent latent nodes $\mathcal{L}_i = \{\ell_{i,1}, \ldots, \ell_{i,k_i}\}$, for some $k_i \geq 1$, each with the same connectivity (i.e. children) as $l_i$. Thus, all told, we distribute $\lambda = \sum_{i \in [K]} k_i$ latents across the cliques, a parameter called the *latent degrees of freedom*. It is easy to check that no matter how the $\lambda$ latent degrees of freedom are distributed, the resulting DAG has the same independence structure over the measurement variables as $\mathcal{G}$. This provides a rigorous device for increasing complexity without affecting the causal structure, and moreover, $\lambda$ is a flexible tuning parameter that can be set arbitrarily large in practice, resulting in potentially overparametrized models. We call the resulting graph a *NCFA-graph* of $\mathcal{G}$ with $\lambda$ latent degrees of freedom.

**Definition 3.3.** Let $\mathcal{G}$ be a minimum MCM graph for the UDG $\mathcal{U} = \langle [n], E \rangle$ and the minimum edge clique cover $\mathcal{C} = \{C_1, \ldots, C_k\}$ of $\mathcal{U}$. A *NCFA graph* of $\mathcal{G}$ with $\lambda$ latent degrees of freedom is a graph $\widetilde{\mathcal{G}}$ with node set $[n] \cup \widetilde{\mathcal{L}}$ and edge set $\widetilde{E}$ where

$$\widetilde{\mathcal{L}} = \mathcal{L}_1 \cup \cdots \cup \mathcal{L}_k \qquad \text{for} \qquad \mathcal{L}_i = \{\ell_{i,1}, \ldots, \ell_{i,k_i}\}, \qquad k_i \geq 1 \ \forall i \in [K],$$

and

$$\widetilde{E} = \{\ell_{i,m} \to j : \forall j \in C_i, \ \forall m \in k_i, \ \forall i \in [K]\}.$$

Each node $\ell_{i,m}$ represents a latent variable $Z_{i,m}$. Since the latent nodes in $\mathcal{L}_i$ all have the same connectivity as the single latent $l_i$, their joint distribution $f(L_i) = \prod_{m=1}^{k_i} f(Z_{i,m})$ represents the common cause of the measurement variables corresponding to the nodes in $C_i$, which was previously only represented by $l_i$ in $\mathcal{G}$. The factors to be inferred from a factor analysis perspective are now the random vectors $L_1, \ldots, L_K$ with $L_i = (Z_{i,1}, \ldots, Z_{i,k_i})$, which still have the causal interpretation afforded by the minimum MCM graph. However, the multiple latents provide us flexibility to model the effects of the causal factors.

**Definition 3.4.** A *NCFA model* is a joint distribution $(M_1, \ldots, M_n)$ for which there is a NCFA-graph $\widetilde{\mathcal{G}} = \langle [n] \cup \widetilde{\mathcal{L}}, \widetilde{E} \rangle$ and functions $f_1, \ldots, f_n$ for which $M_i := f_i(\mathrm{pa}_Z(i), \epsilon_i)$ for all $i \in [n]$, where $\mathrm{pa}_Z(i) := \{Z_{j,m} : \ell_{j,m} \in \mathrm{pa}_{\widetilde{G}}(i)\}$.

When modeling a distribution via a NCFA model, the functions $f_i$ are treated as unknowns to be inferred via a deep generative model such as a VAE. The encoder maps the observations into the latent space as the joint posterior distribution $f(Z|M_1, \ldots, M_n)$ where $Z$ is the random vector that collects the $Z_{j,m}$, and the decoder maps latents to observations according to the factorization

$$f(M_1, \ldots, M_n | Z) = \prod_{i=1}^{n} f(M_i | \mathrm{pa}_Z(i)).$$

The joint distribution of the latent space is $f(Z) = \prod_{i=1}^{K} f(L_i)$; i.e., it is a product of the (joint) distributions we have specified to represent each of the latent common causes in the minimum MCM model $\mathcal{G}$ for $\mathcal{U}$. Following training of the VAE, the model may be used to generate predictions in the observation space via draws from the latent space. Since our representation of the latent space was constructed according to the minimum MCM graph $\mathcal{G}$, the resulting predictions can be viewed as causally informed; i.e., they are observations generated from the estimated distribution of the latent primary causes of the measurement variables.

## 3.2 Identifiability of minimum MCM graphs and ECC-model equivalence

While the UDG is identifiable, there may exist multiple minimum MCM graphs that yield the same UDG. This is because an undirected graph may have multiple, distinct minimum edge clique covers (see, for instance, the example provided in Appendix B). In other words, similar to DAGs, minimum MCM graphs may be equivalent when provided with only observational data.

**Definition 3.5.** We say that two minimum MCM graphs $\mathcal{G} = \langle [n] \cup L, E \rangle$ and $\mathcal{G}' = \langle [n] \cup L', E' \rangle$ are *ECC-observationally equivalent* if $i$ and $j$ are $d$-separated given $\emptyset$ in $\mathcal{G}$ if and only if they are $d$-separated given $\emptyset$ in $\mathcal{G}'$.

While there exist equivalence classes of minimum MCM graphs containing multiple elements, there also exist classes that are singletons; in other words, there exist undirected graphs (UDGs) with a unique minimum edge clique cover. For such UDGs, the minimum MCM graph is identifiable.

---

**Algorithm 1:** Neuro-Causal Factor Analysis (NCFA)

> **input** : sample $S$ of measurement variables $M$
> **parameter** : significance level $\alpha$, latent degrees of freedom $\lambda$
> **output** : neuro-causal factor model $\langle \widetilde{\mathcal{G}}, f_{[n]}, \epsilon \rangle$, with NCFA graph $\widetilde{\mathcal{G}}$, loading functions $f_{[n]}$, and residual measurement errors $\epsilon$

1 Estimate $\mathcal{U}$, the undirected dependence graph, via pairwise marginal independence tests with threshold given by $\alpha$;

2 Identify a minimum edge clique cover $\mathcal{C}$ of $\mathcal{U}$ and construct the corresponding minimum MCM graph $\mathcal{G}$;

3 Assign the remaining $\lambda - |\mathcal{C}|$ latents to the cliques in $\mathcal{C}$ to produce the NCFA-graph $\widetilde{\mathcal{G}}$;

4 Estimate functions $f_{[n]}$ using a VAE constrained by $\widetilde{\mathcal{G}}$, with residual measurement errors $\epsilon$;

5 **return** $\langle \widetilde{\mathcal{G}}, f_{[n]}, \epsilon \rangle$

---

**Theorem 3.6.** *Suppose that the data-generating distribution is Markov to a minimum MCM graph $\mathcal{G}$. Then the DAG $\mathcal{G}$ is identifiable from the data-generating distribution if:*

1. *The UDG $\mathcal{U}$ for $\mathcal{G}$ admits a unique minimum edge clique cover, and*

2. $M_i \perp\!\!\!\perp M_j \iff i - j \notin E^{\mathcal{U}}$.

**Corollary 3.7.** *Suppose that the data-generating distribution is Markov to a minimum MCM graph $\mathcal{G}$ satisfying the 1-pure-child assumption, namely, for each latent $l_i$ in $\mathcal{G}$ there exists a measurement node $i^*$ such that $\mathrm{pa}_{\mathcal{G}}(i^*) = \{l_i\}$. Then $\mathcal{G}$ is identifiable.*

Proofs are deferred to Appendix B. The identifiability result in Corollary 3.7 applies to models that are of practical interest (e.g. as in Donoho & Stodden, 2003; Arora et al., 2012; Bing et al., 2020; Moran et al., 2023). However, Theorem 3.6 shows that these are not the only models to which the identifiability result applies. An example of a UDG that admits a unique minimum edge clique cover but does not satisfy the pure measurement variable condition is given in Appendix B.

## 4 NEURO-CAUSAL FACTOR ANALYSIS

We now present our main contribution, the Neuro-Causal Factor Analysis (NCFA) algorithm, given in Algorithm 1. The NCFA algorithm runs by the logic described in Section 3: namely, it infers a UDG from data, identifies a minimum edge clique cover $\mathcal{C} = \{C_1, \ldots, C_K\}$ for $\mathcal{U}$, builds the corresponding NCFA-graph $\mathcal{G}$ with $\lambda$ latent degrees of freedom and then trains a VAE according to the functional relationships among the measurement and latent variables specified by $\widetilde{\mathcal{G}}$.

To estimate the UDG, pairwise marginal independence tests are performed. Starting with the complete graph, the edge $i - j$ is removed whenever $M_i$ and $M_j$ are deemed independent, i.e. according to a test with statistics such as distance-covariance (Székely et al., 2007; Markham et al., 2022) or Chatterjee's coefficient (Chatterjee, 2021; Lin & Han, 2022). A minimum edge clique cover is then identified for the estimated UDG $\widehat{\mathcal{U}}$. In general, this is an NP-hard problem, however there are both exact algorithms that work well for small graphs and heuristic algorithms that scale to large graphs (Gramm et al., 2009; Conte et al., 2020; Ullah, 2022).

Once a minimum edge clique cover is identified, the corresponding NCFA graph with $\lambda$ latent degrees of freedom is constructed. Here, we ensure that at every clique in the minimum edge clique cover of $\widehat{\mathcal{U}}$ is assigned at least one latent variable. The remaining $\lambda - K$ latents are then distributed uniformly over the cliques. In this implementation of NCFA, we set default $\lambda = \lfloor n^2/4 \rfloor$, a known upper bound on the number of cliques in a minimum edge clique cover of a graph on $n$ nodes (Erdős et al., 1966). Finally, a VAE for the functional relations specified by the NCFA-graph is trained. One could, in principle, alternatively use any deep generative model. See Appendix C for further details.

Since NCFA constructs its model via the MCM graph $\widehat{\mathcal{U}}$, the estimated factors (i.e., joint distributions) $f(L_i)$ in the factorization of the latent distribution represent the distributions for the primary causes of the measurement variables to which the latent nodes in $\mathcal{L}_i$ are connected. This yields a factor

analysis model in which the latent factors can justifiably be causally interpreted. Furthermore, while each latent variable $Z_{i,j}$ is assigned a Gaussian prior in the VAE, by assigning $\mathcal{L}_i = \{\ell_{i,1}, \ldots, \ell_{i,k_i}\}$ latents to each clique $C_i$, instead of a single latent $l_i$, each causal latent in the minimum MCM graph is modeled as a mixture distribution which can be arbitrarily non-Gaussian. Hence, the estimated factors have both a causal interpretation while additionally being as nonlinear as necessary.

## 5 APPLICATIONS ON SYNTHETIC AND REAL DATA

We now present results of applying NCFA to synthetic and real data sets, observing that the performance of NCFA is competitive with classical VAEs while additionally offering a nonlinear, causally interpretable factor model. We provide a Python implementation of the NCFA algorithm as well as scripts for reproducing all of the following results, released as a free/libre software package: https://after.review. Here we summarize our main findings; the full experimental protocol and details can be found in the appendix, including details on the NCFA implementation (Appendix C), evaluation metrics (Appendix D), synthetic data generation and additional results (Appendix E), and additional results on real data (Appendix F).

NCFA faces a trade-off between causal constraints and expressivity: an unconstrained, fully connected VAE ignores this structure, and has free reign to fit the data arbitrarily, at the cost of interpretability and potentially *a*causal relationships (e.g. spurious correlations). The additional structure offered by the minimum MCM graph in NCFA brings in causal structure and interpretation, but can hamper training if the structure is incorrect. Of course, when the causal structure is correct, there should be no significant loss in expressivity. Thus, ideally we will see no significant degradation in the loss, which is an indicator of structural fidelity. We measure this with the metric $\Delta$ which is the difference between the loss of an unconstrained, baseline VAE and the NCFA loss. On synthetic data where we know the causal ground truth, we can also directly measure structural fidelity using graph comparison metrics. See Appendix D for detailed definitions of our metrics.

Except for the last experiment, no hyperparameter tuning was performed, and instead default, reasonable choices are used (e.g. $\alpha = 0.05$ and $\lambda = \lfloor n^2/4 \rfloor$). We anticipate improvements are possible with careful hyperparameter tuning.

**Synthetic data** We summarize some key results on the synthetic data, compared to both a ground truth causal model and a baseline VAE, in Figure 3. Results are grouped according to edge density of the generating UDG, shown along the $x$-axis. Figure 3a contains box plots of distance between the true MCM causal structure and that learned by NCFA (lower is better). Here, distance between MCM graphs is measured using the *Structural Frobenius Difference* (SFD), which is a modification of the more common Structural Hamming Distance (SHD) for graphs with possibly different numbers of nodes (see Appendix D for more details on SFD and its relation to SHD). Figure 3b contains box plots of Validation-$\Delta$, the difference between the final validation loss of the baseline VAE and that of NCFA (higher is better). Additionally, we report that NCFA learned the exact true causal structure at a proportion of 0.91 for density $p = 0.1$, at 0.56 for $p = 0.2$ and between 0.39 and 0.43 for other values of $p$.

As is commonly seen in causal discovery tasks, NCFA recovers causal structure well in the sparse setting but increasingly less so in denser settings. Causal discovery is notoriously difficult, especially in the small-sample regime, but NCFA benefits from only needing to perform *marginal* independence tests (so the conditioning set is always empty). In terms of performance as a generative model, we see that NCFA generally improves the validation loss compared to the baseline VAE since the median loss difference is above $0$ for all edge densities except for $p = 0.1$, even as the true graph density increases. This indicates both that the causal structure provides helpful constraints in the NCFA pipeline and that NCFA is robust in the face of moderate misestimation of the causal structure.

**Real data** We ran NCFA on two real datasets, MNIST and TCGA, comparing its performance to a baseline VAE. In both cases, there is no ground truth causal graph, so we focus on VAE metrics as a benchmark. We report the results in Table 1. For MNIST, sample size is much larger than number of measurement variables $n$, but this is not true of TCGA. When run using default settings for $\alpha, \lambda$ in the first two rows, we see that NCFA achieves comparable training and validation to the baseline VAE, demonstrating that it learns reasonable constraints (i.e. causal relations) as well as its ability

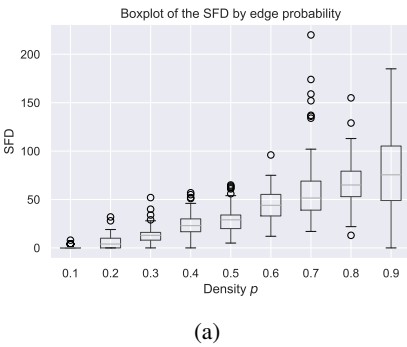 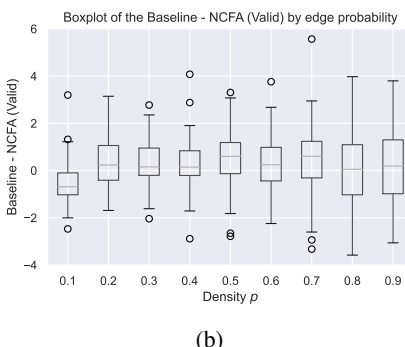

(a)                                                        (b)

Figure 3: Results of NCFA on synthetic data sets from randomly generated graphs: (a) shows distance (SFD) between learned causal structures and the ground truth; (b) shows Validation-$\Delta$, the difference of validation loss between baseline VAE and NCFA (higher means better performance for NCFA).

to scale well to high-dimensional settings. In fact, for TCGA the training and validation losses are lower for NCFA, suggesting that incorporating the causal structure learned by NCFA improved model performance. Curiously, for MNIST, the minimum MCM graph consisted of just a single latent (i.e., $|L| = 1$), suggesting the causal structure in this dataset is limited, which matches expectations. This does not mean that there are not multiple, interpretable latents to be discovered as is well-documented in the literature, but perhaps that these latents do not have a strong causal interpretation.

Table 1: Results of NCFA on two real data sets

|        | samp size | $n$  | $\alpha$ | $\lambda$ | $|L|$ | Training-$\Delta$ | Validation-$\Delta$ |
|--------|-----------|------|----------|-----------|-------|-------------------|---------------------|
| MNIST  | 42000     | 784  | 0.05     | 153664    | 1     | -0.00475          | -0.04814            |
| TCGA   | 632       | 1000 | 0.05     | 250000    | 8129  | 0.11488           | 0.11865             |
| MNIST  | 42000     | 784  | 0.001    | 7800      | 560   | -76.682           | -74.163             |
| TCGA   | 632       | 1000 | 0.05     | 10000     | 969   | -78.721           | -68.117             |

On both datasets, the default $\lambda$ and maximum allowed $|L| < \lambda$ were quite large, so we also ran experiments under the 1-pure-child assumption (see Appendix F for details), which guarantees that $|L| \leq n$, allowing us to safely reduce $\lambda$ from $\lfloor n^2/4 \rfloor$ to, e.g., $10n$. Additionally, we decreased $\alpha$ to $0.001$ for MNIST, taking advantage of the large sample size and encouraging NCFA to learn a sparser structure. However, based on the training and validation differences, NCFA failed to converge properly compared to the baseline VAE. In the case of MNIST, we attribute this to it arguably being a data set without causally meaningful sparse latents. For TCGA, the performance of NCFA without the 1-pure-child assumption yielded a better performance than the baseline VAE. Hence, the decrease in performance of NCFA under this constraint could suggest that the true causal structure of TCGA simply does not abide by the 1-pure-child assumption. Collectively, these results suggest that NCFA with default parameter specifications appears to yield competitive, if not improved, performance over baseline VAE models that successfully incorporate causal structure when it is present to be learned. When NCFA has free reign to learn whatever causal structure (when it exists, as in TCGA) can be gleaned from the data, it appears to benefit training. However, the second round of experiments suggest that one should take care when adjusting the algorithm to fit a specified causal structure, such as the 1-pure-child constraint, as forcing possibly nonexistent causal structure into the model may be detrimental to the models predictive capabilities. This is in line with the observation at the start of Section 5 that one risks hampering training when the causal structure is misspecified.

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

# A    MeDIL Causal Models

The NCFA models used in this paper are a subfamily of the models known as MeDIL causal models, originally introduced in (Markham & Grosse-Wentrup, 2020). For contextualization purposes, we give a brief description of MeDIL causal models here.

A *MCM graph* is a triple $\mathcal{G} = \langle M, L, E \rangle$ where $M = \{1, \ldots, n\}$ and $L = \{l_1, \ldots, l_K\}$ are disjoint sets of vertices corresponding, respectively, to observed variables and latent variables, and $E$ is the collection of directed edges between the nodes. In a MCM graph we require that the nodes in $M$ are all *sinks* (i.e., have out-degree 0) and that the edges $E$ are such that $\mathcal{G}$ is a *directed acyclic graph* (DAG). A distribution belongs to the *Measurement Dependence Inducing Latent (MeDIL) Causal Model* if it factors according to $\mathcal{G}$; i.e., the probability density function (or probability mass function) satisfies

$$f(x_1, \ldots, x_n, x_{l_1}, \ldots, x_{l_K}) = \prod_{i=1}^{n} f(x_i | x_{\mathrm{pa}_{\mathcal{G}}(i)}) \prod_{j=1}^{K} f(x_{l_j} | x_{\mathrm{pa}_{\mathcal{G}}(l_i)}).$$

Since we do not observe the latent variables, but only the observed variables, $X_1, \ldots, X_n$ the distributions of interest factorizes as

$$f(x_1, \ldots, x_n) = \int_{\mathcal{X}_L} \prod_{i=1}^{n} f(x_i | x_{\mathrm{pa}_{\mathcal{G}}(i)}) \prod_{j=1}^{K} f(x_{l_j} | x_{\mathrm{pa}_{\mathcal{G}}(l_i)}) dx_L = \prod_{i=1}^{n} \varphi_i(x_i). \tag{1}$$

Note that since all observed variables are sink nodes in $\mathcal{G}$, integration over the latents yields a factorization of the distribution into a product of potential functions, one for each observed variable $X_i$ that depends only on $x_i$. The MeDIL model $\mathcal{M}(\mathcal{G})$ consists of all observable distributions that arise according to this factorization.

In general, MCM graphs may have a complex directed acyclic structure over the latents that induce correlations amongst the observables. However, since we only consider the observed variables, we may reduce to a minimum MCM representation of the distribution. In a *minimum MCM graph*, we further assume that the latent variables are *source nodes* (i.e., have in-degree 0) and out-degree at least 1. Hence, a *minimum MeDIL model*; i.e., a MeDIL model that factorizes as in equation (1) explains the associations amongst the observed variables in the simplest possible way: via a collection of independent latents. These independent latents can be thought of as source nodes in the original MCM graph $G$, and the associated minimum MCM graph as the graph produced after we marginalize out all non-source latent variables. While learning the complex causal structure on the latents in a general MCM graph may be intractable, one can apply existing methods to estimate the simpler minimum MCM structure. Intuitively, one can think of learning the minimum MCM graph of a model as identification of the "primary causes" of the associations between the observed variables.

It follows from the factorization of a distribution according to a minimum MCM graph that the only conditional independence constraints amongst the observed variables are marginal independence constraints. These, in turn, encode the condition that two nodes do not have a shared (latent) parent. From the causal perspective, we see that two nodes in the system are marginally independent if and only if they share a latent common cause. Hence, minimum MeDIL models are the natural representation of Reichenbach's Common Cause Principle.

It follows that a natural representation of the model using only the observed variables is via an undirected graph $\mathcal{U} = \langle M, E \rangle$, where $i - j \notin E$ if and only if $i$ and $j$ are independent in the joint distribution of the observed variables. We call this graph the *unconditional dependence graph* (UDG) of the model. A *minimum edge clique cover* of the undirected graph $U$ is a collection of cliques $\mathcal{C} = \{C_1, \ldots, C_K\}$ for which every pair $i, j$ satisfying $i - j \in E$ is contained in at least one clique in $\mathcal{C}$ and there exists no set of cliques with this property that has cardinality smaller than $|\mathcal{C}|$. Since a UDG depends only on the observed variables, it is possible to learn a UDG from the available data via pairwise marginal independence tests. A minimum MCM graph that captures the marginal independence structure encoded via the UDG is then the minimum MCM graph where we have one latent $l_i$ for each clique $C_i \in \mathcal{C}$ and $l_i$ has a directed arrow to each node $j \in C_i$ (and no other adjacencies).

To learn the assignment of latents producing a minimum MCM graph from a UDG $\mathcal{U}$, we must learn a minimum edge clique cover of $\mathcal{U}$. The problem of identifying a minimum edge clique cover of an

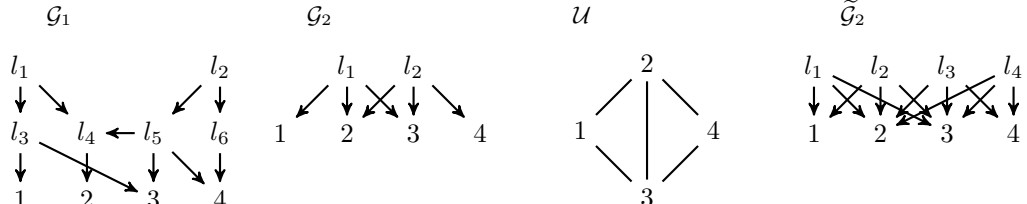

Figure 4: A MCM graph $\mathcal{G}_1$ with observed variables $1, 2, 3, 4$, a minimum MCM graph $\mathcal{G}_2$ representing the same set of observable distributions $f(x_1, x_2, x_3, x_4)$ and the associated UDG $\mathcal{U}$. The (unique) minimum edge clique cover of $\mathcal{U}$ is $\mathcal{C} = \{\{1, 2, 3\}, \{2, 3, 4\}\}$. Hence, $\mathcal{G}_2$ has exactly one latent variable for each of these two cliques, connected to exactly the nodes in a single clique in $\mathcal{C}$. The NCFA-graph $\widetilde{\mathcal{G}}_2$ is an augmentation of $\mathcal{G}_2$ for $\lambda = 4$ latent degrees of freedom (see Section 3). All four graphs encode a single marginal independence constraint, $X_1 \perp\!\!\!\perp X_4$ (shown via the $d$-separation of 1 and 4 in $\mathcal{G}_1, \mathcal{G}_2$, and $\widetilde{\mathcal{G}}_2$, and the lack of edge $1 - 4$ in $\mathcal{U}$), and hence represent the same family of observed distributions.

undirected graph is NP-hard. There are exact algorithms (Gramm et al., 2009; Ullah, 2022) showing it to be NP-complete and fixed-parameter tractable. However, there are also polynomial-time heuristic algorithms (Conte et al., 2020) that can efficiently handle very large graphs.

# B  IDENTIFIABILITY OF MINIMUM MCM GRAPHS: EXAMPLES AND COUNTEREXAMPLES

*Proof of Theorem 3.6.* Because the data-generating distribution is Markov to $\mathcal{G}$, the use of any consistent hypothesis test for independence (e.g., (Chatterjee, 2021)) is guaranteed to identify all marginal independence statements. Using the UDG $U$ induced by these marginal independence statements (according to 2), and using (Markham & Grosse-Wentrup, 2020, Proposition 7) to associate $\mathcal{G}$ to a minimum edge clique cover, the uniqueness of the edge clique cover (according to 1) guarantees the identifiability of $\mathcal{G}$. $\square$

*Proof of Corollary 3.7.* The association of a minimum MCM graph $\mathcal{G}$ with a minimum edge clique cover of $U$ (Markham & Grosse-Wentrup, 2020, Proposition 7) induces an association between a maximum independent set in $U$ and a set of pure children in $\mathcal{G}$ with different parents. Observe that $G$ satisfies the 1-pure-child assumption if and only if there exists a maximum independent set $\{i^*\}_{i=1}^K$ in $U$ consisting of one pure child for each latent. In other words, $G$ satisfies the 1-pure-child assumption if and only if the independence number (i.e., size of a maximum independent set, which is the number latents with at least one pure child) and intersection number (i.e., the number of cliques in a minimum edge clique cover, which is the number of latents) of its corresponding $U$ are equal. Such graphs are known (Deligeorgaki et al., 2023, Lemma 2.3.5) to have a unique minimum edge clique cover, so $\mathcal{G}$ is identifiable by Theorem 3.6. $\square$

An example of a minimum MCM graph that cannot be identified because it admits multiple edge clique covers is the graph:

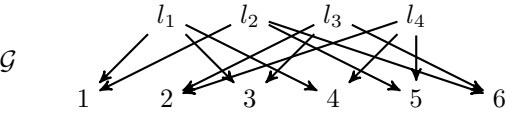

The UDG of $\mathcal{G}$ is the edge graph $\mathcal{U}$ of the octahedron which admits exactly two minimum edge clique covers: $\mathcal{C}_1 = \{\{1, 3, 4\}, \{1, 5, 6\}, \{2, 3, 6\}, \{2, 4, 5\}\}$ and $\mathcal{C}_2 = \{\{1, 3, 6\}, \{1, 4, 5\}, \{2, 3, 4\}, \{2, 5, 6\}\}$. $\mathcal{C}_1$ recovers the minimum MCM graph $\mathcal{G}$, whereas $\mathcal{C}_2$ produces the minimum MCM graph $\mathcal{H}$, depicted below:

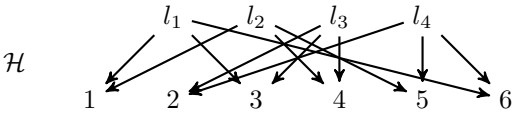

This issue arises due to the spherical structure of the UDG $\mathcal{U}$, depicted on the left below. In the remaining two figures, the each shaded triangle represents the children of a latent node (i.e., a clique in a minimum edge clique cover) of a possible MCM graph with associated UDG $\mathcal{U}$. The middle figure represents the minimum MCM graph $\mathcal{G}$ and the rightmost figure represents the alternative $\mathcal{H}$.

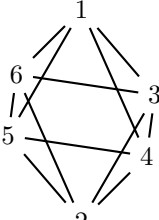
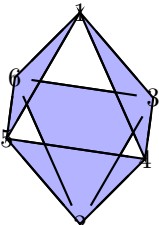
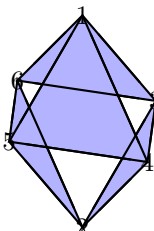

The spherical structure of this graph creates symmetry allowing for multiple minimum edge clique covers. In particular, this gives an example to where we lack identifiability and Theorem 3.6 does not apply. Interestingly, we see that this lack of identifability can be "worst possible" in the sense that if the true minimum MCM graph is $\mathcal{G}$, but the wrong edge clique cover is chosen, we could learn $\mathcal{H}$, which captures none of the true latents and instead specifies four completely incorrect latents. Future work, possibly incorporating interventional knowledge, to address such identifiability concerns would be of interest.

On the other hand, graphs admitting a *pure measurement variable* in each clique of a minimum edge clique cover will have a unique minimum edge clique cover, making the resulting minimum MCM graph identifiable (see Theorem 3.6). From a graphical perspective, we call this condition the *1-pure child constraint* since it insists that each latent node $l_i$ in the minimum MCM graph has (at least) one child $i^*$ which has no other parents than $l_i$ (i.e., $l_i \to i^*$ is an edge of $\mathcal{G}$ and $i^*$ has no other adjacencies in $\mathcal{G}$). However, the 1-pure-child constraint is not necessary for a UDG to have a unique minimum edge clique cover and hence for identifiability to hold (according to Theorem 3.6). In particular, the following UDG $\mathcal{U}$ has the unique minimum edge clique cover $\mathcal{C} = \{\{1,4\},\{2,5\},\{3,6\},\{4,5,6\}\}$ and hence the corresponding minimum MCM graph $\mathcal{G}$ is identifiable:

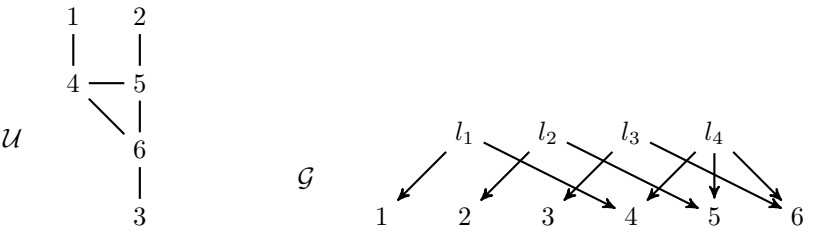

Note here that the latent variables $l_1, l_2$ and $l_3$ each have a pure child ($1, 2$ and $3$, respectively). However, the latent $l_4$ does not. Hence $\mathcal{G}$ is identifiable but does not satisfy the 1-pure-child constraint.

## C  NCFA IMPLEMENTATION

We provide a Python implementation (including thorough documentation) of the NCFA algorithm (Algorithm 1) as well as scripts for reproducing all of our results, released as a free/libre software package: `https://after.review`.

Our implementation makes use of the following Python packages: `NumPy` (Harris et al., 2020), `PyTorch` (Paszke et al., 2019), `dcor` (Ramos-Carreño & Torrecilla, 2023), and `xicorrelation`

(https://github.com/jettify/xicorrelation). Additionally, we provide a Python wrapper of the Java `ECC` package (https://github.com/Pronte/ECC).

Further details can be found in our code and documentation, but we summarize the most important implementation details in the following:

- **Marginal independence testing (Step 1, Algorithm 1)**: We use statistical hypothesis tests of independence (either using `dcor` for distance covariance based tests or `xicorrelation` for Chatterjee's coefficient based tests) for specified threshold value $\alpha$, with the $p$-values being computed based on the asymptotic theory as opposed to with permutation tests.

- **Edge clique cover (Step 2, Algorithm 1)**: We use the default heuristic solver of the `ECC` package, except when making the 1-pure-child assumption, in which case the corresponding learned UDG $\widehat{\mathcal{U}}$ has equal intersection and independence numbers, a condition known (Deligeorgaki et al., 2023) to allow for a subcubic time exact solver, which we implement ourself (see package documentation for further details).

- **Assignment of latent degrees of freedom according to $\lambda$ (Step 3, Algorithm 1)**: First, each of the $K$ cliques from the edge clique cover is assigned one latent; then, the remaining $\lambda - K$ latents are distributed equally over the cliques.

- **VAE architecture (Step 4, Algorithm 1)**: The learned UDG $\widehat{\mathcal{U}}$ induces an NCFA model with NCFA graph $\widehat{\mathcal{G}}$ (cf. Section 3.1). The NCFA graph encodes the structural connections between the latents and observables as a biadjacency matrix, which is used to mask the connections in the decoder of the VAE. In this way, each observable only depends on the latent it is connected to in the learned NCFA graph $\widehat{\mathcal{G}}$ (cf. Definition 3.4). The decoder structure is given by a single layer of the SparseLinear network with a masked linear activation function. The model was trained using the AdamW optimizer with a learning rate of 1e-5 and each time run for 200 epochs.

All experiments were run on Intel(R) Xeon(R) Gold 6154 CPU @ 3.00GHz. Total combined runtime was about 50 hours.

## D    EVALUATION METRICS

**Distance measures**    Unlike classical VAE methods, NCFA has a causal discovery step in which it infers the UDG upon which the VAE in the model is based. The structural Hamming distance (SHD) is commonly used to quantify the difference between graph structures and is simply defined to be the number of edges that appear in one graph but not the other. SHD, however, is not applicable in our case as (i) it is defined for graphs having the same number of vertices, whereas for a fixed finite sample from a set of measurement variables, differently estimated minimum MCMs may have differing numbers of latent variables, and (ii) even if we compute, e.g., the SHD between the undirected graphs $\mathcal{U}, \mathcal{U}'$ (which *do* have the same number of vertices), this aligns poorly with intuitions about distance between their respective generating MCM graphs $\mathcal{G}, \mathcal{G}'$. To remedy this, we introduce the following distance:

**Definition D.1.** The *structural Frobenius* (or *medil*) *distance* between biadjacency matrices $B_1 \in \{0,1\}^{K_1,n}$ and $B_2 \in \{0,1\}^{K_2,n}$, corresponding to minimum MCM graphs $\mathcal{G}_1, \mathcal{G}_2$ each with $n = |M|$ measurement variables and $K_i$ latents, respectively, for $i = 1, 2$, is defined as

$$\text{SFD}(B_1, B_2) \coloneqq \|B_1^\top B_1 - B_2^\top B_2\|_F^2.$$

Note that, by virtue of being defined using the Frobenius norm, the medil distance is a proper distance metric. The intuition behind this definition is that we transform a given minimum MCM structure into a weighted undirected graph, allowing it to be easily compared to other minimum MCM structures over the same set of measurement variables. Via this transformation, the weights keep track of how many latent parents each pair of measurement variables have in common as the $(i, j)$-th entry in the matrix $B_1^\top B_1$ records the number of colliderless paths between measurement variables $i$ and $j$ in the graph. For our graphs, two nodes admit such a path between them if and only if they are connected by a latent, and this connection adds exactly 1 to the $(i, j)$-th matrix entry. Hence, these matrices will

be equal if and only if the MeDIL models are identical (for a consistent ordering of measurement variables) (Deligeorgaki et al., 2023, Section 2.2). Figure 5 provides an example comparing the SFD between minimum MCMs and the SHD between their corresponding (unweighted) undirected graphs.

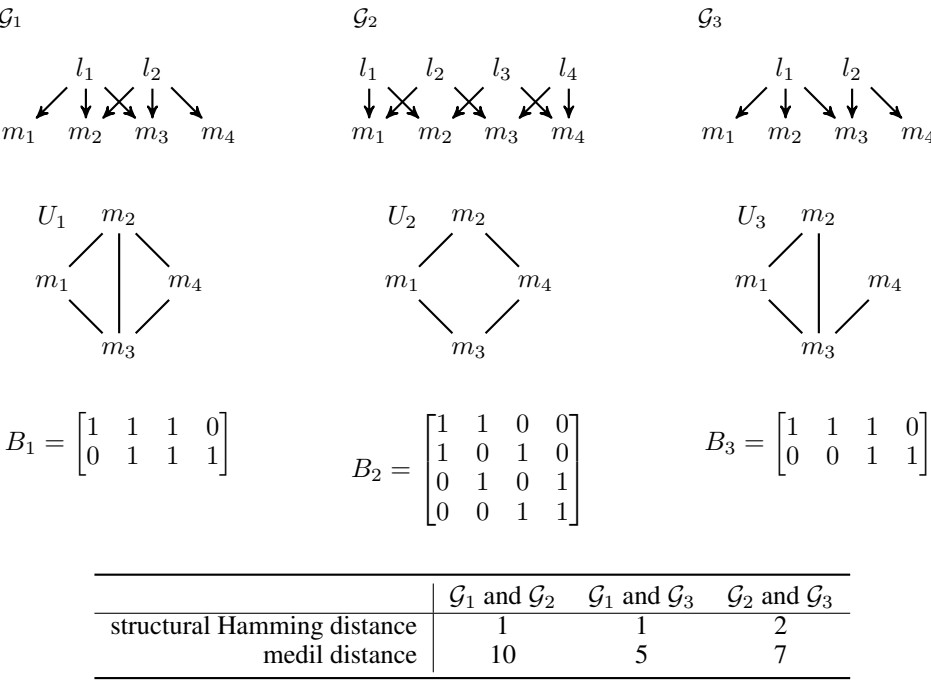

| | $\mathcal{G}_1$ and $\mathcal{G}_2$ | $\mathcal{G}_1$ and $\mathcal{G}_3$ | $\mathcal{G}_2$ and $\mathcal{G}_3$ |
|---|---|---|---|
| structural Hamming distance | 1 | 1 | 2 |
| medil distance | 10 | 5 | 7 |

Figure 5: Pairwise comparison of structural Hamming distance and medil distance on three causal factor structures, $\mathcal{G}_1, \mathcal{G}_2$, and $\mathcal{G}_3$. For each structure $\mathcal{G}_i$, we also show its associated undirected graph $U_i$ and its biadjacency matrix $B_i$.

**VAE loss function** The training and validation losses reported here and in the main paper are standard ELBO metrics (e.g. Kingma & Welling, 2014) computed using a 70/30 training/validation split.

## E  SIMULATIONS: SPECIFICATIONS AND ADDITIONAL RESULTS

**Data Generation** We generated 10 Erdős-Rényi random undirected graphs (Gilbert, 1959) on 10 nodes for each density value $p \in \{0.1, 0.2, \ldots, 0.9\}$. These are actual density values, not expectations, i.e., for each density value we sample uniformly from graphs with $p\binom{10}{2}$ edges. These random graphs constitute UDGs, so we compute their minimum edge clique covers to determine the corresponding MCM structure. Importantly, while this approach lets us sample from the entire space of minimum MCM structures, it only lets us directly specify the graph density of the UDG over measurement variables and not the number of latent causal factors or the density of the MCM structure.

Using the ground truth MCM structures as generated above, we construct a linear factor model (Definition 2.1) with factor loading weights drawn from the interval $[-2, -0.5) \cup (0.5, 2]$ and standard normal error terms, from which we draw 10 different data sets (per edge density $p$) containing 1000 observations each, totaling 90 different ground truth causal structures and 900 different data sets.

**Model Hyperparameters** For each data set, we trained a NCFA model with $\alpha = 0.05, \lambda = 25$ (which includes estimating the causal structure) as well as two baselines: (i) a classic fully connected VAE with the same number of latent degrees of freedom, and (ii) a NCFA model given the true causal structure and with the latent degrees of freedom $\lambda$ being set equal to the true number of causal latent factors $K = |L|$.

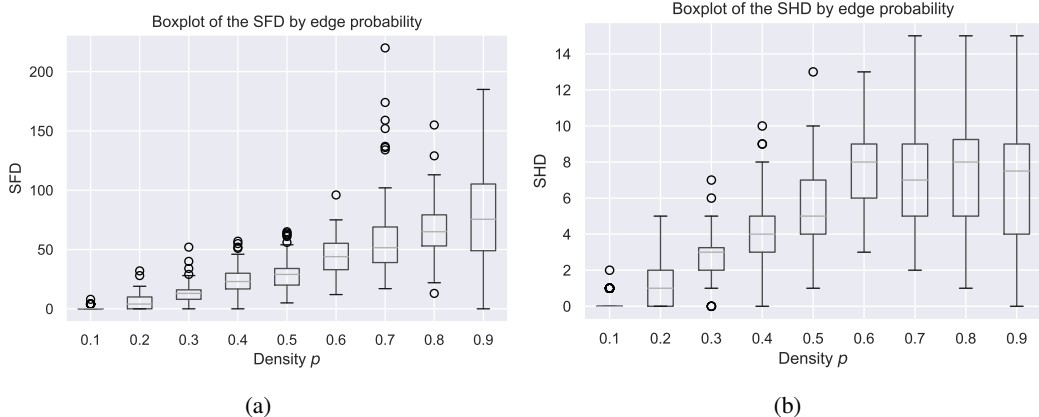

(a)                                                          (b)

Figure 6: Results of NCFA on synthetic data sets from randomly generated graphs, in terms of causal structure learning: (a) shows structural Frobenius distance (SFD) between learned (biadjacency matrix representation of) causal structures and the ground truth; (b) shows structural Hamming distance (SHD) between (undirected graph representation of) learned causal structures and the ground truth.

**Additional Results**    Figure 6 shows the distances (both SFD and SHD) between the learned and true causal structures, in the form of a box plot for each different edge density $p$, demonstrating better performance (in terms of causal structure learning) for sparser graphs. However, in terms of performance as a generative model, Figures 7 and 8 (respectively showing boxplot summary of the $\Delta$ losses and averaged $\Delta$ losses for different generating edge densities) demonstrate that NCFA generally achieves training loss comparable to the baseline VAE while improving the validation loss, and at the same time improving both training and validation loss compared to the ground truth model (owing to NCFA's more expressive VAE architecture and higher $\lambda$). Together, these results indicate both that the causal structure provides helpful constraints in the NCFA pipeline and that NCFA is robust in the face of moderate misestimation of the causal structure.

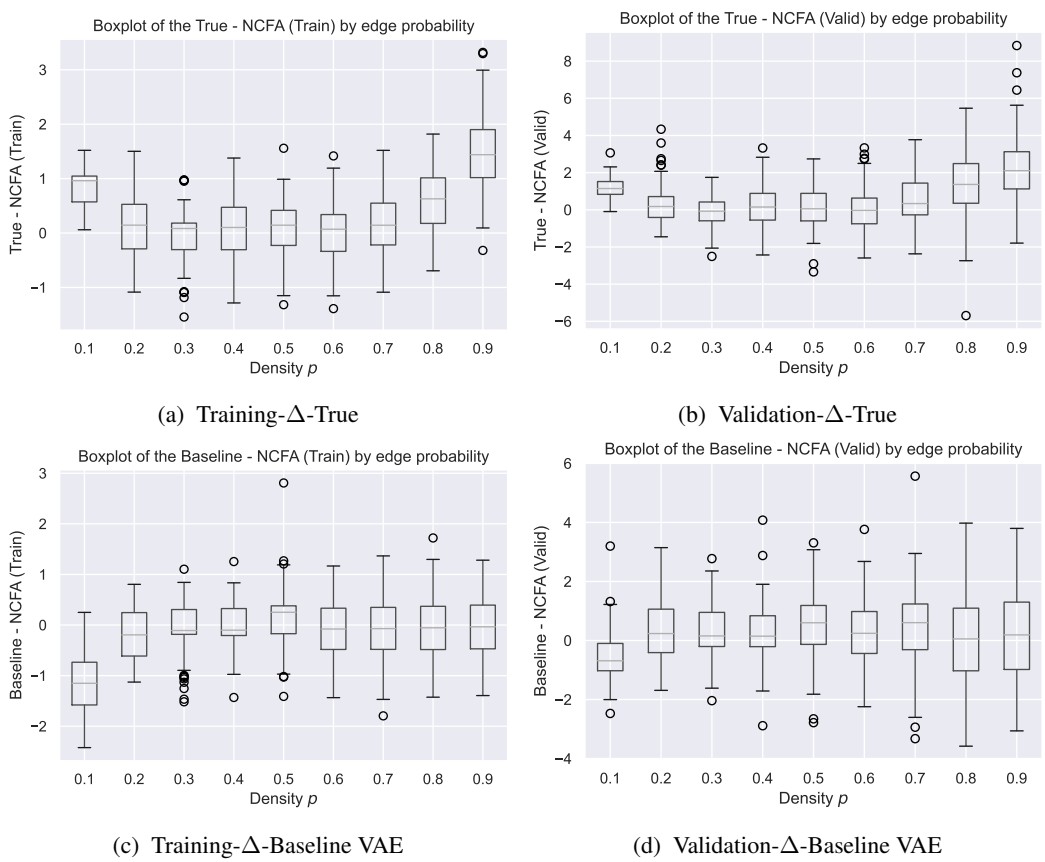

Figure 7: Results of NCFA on synthetic data sets from randomly generated graphs, in terms of final loss of the trained deep generative models: (a) shows Training-$\Delta$-True, the difference of training loss between the learned NCFA model and the NCFA model when given the ground truth causal structure; (b) shows Validation-$\Delta$-True, difference between NCFA and ground truth validation losses; (c) shows Training-$\Delta$-Baseline, difference between NCFA and Baseline VAE training losses; (d) shows Validation-$\Delta$-Baseline, difference between NCFA and Baseline validation losses.

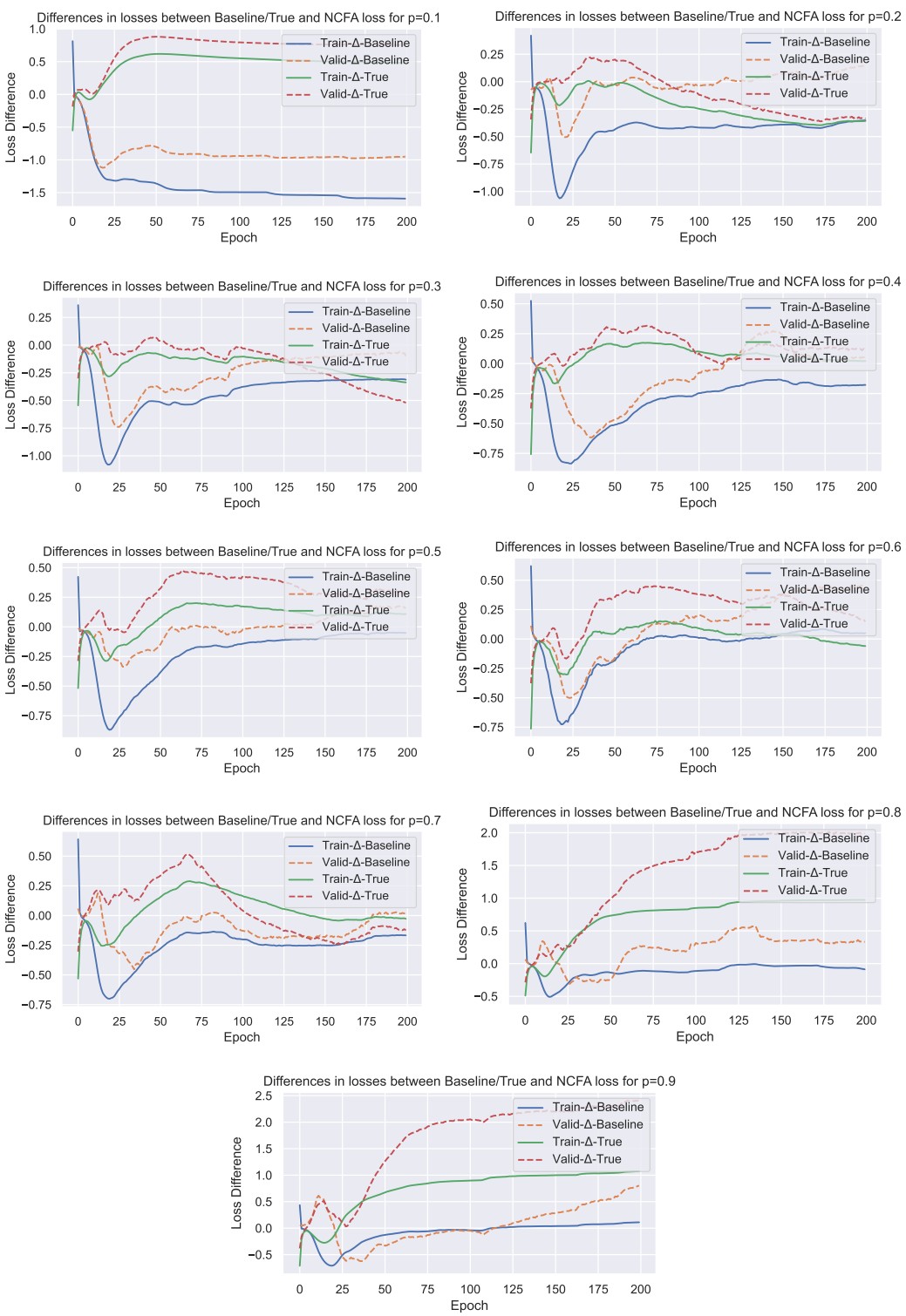

Figure 8: $\Delta$ curves for NCFA versus baseline and ground truth on synthetic data sets from randomly generated graphs, averaged over the 100 data sets for each edge density $p \in (0.1, 0.2, \ldots, 0.9)$.

# F  REAL DATA ANALYSIS: SPECIFICATIONS AND ADDITIONAL RESULTS

**Model Hyperparameters**   As noted in Section 5, we used default settings for the first run on MNIST and TCGA (first and second rows of Table 1, and Figures 9a and 9b ). For the second run on each (third and fourth rows of Table 1, and Figures 9c and 9d ), we tuned the hyperparameters slightly, changing $\alpha = 0.001$ and using distance covariance based tests [3] on MNIST to encourage greater sparsity in the learned NCFA model. Furthermore, for the second run we made the 1-pure-child assumption for both MNIST and TCGA, allowing us to greatly reduce $\lambda$ and make use of a polynomial time exact minimum edge clique cover solver (described in Appendix C).

**Additional Results**   Figure 9 shows the $\Delta$ curves corresponding to the runs described in Section 5 and Table 1, demonstrating that NCFA with default parameter specifications yields competitive, if not improved, performance over baseline VAE models. When NCFA has free reign (as in the default run) to learn whatever causal structure (when it exists, as in TCGA) can be gleaned from the data, it appears to benefit training. However, the second round of experiments suggest that one should take care when adjusting the algorithm to fit a specified causal structure, such as the 1-pure-child constraint, as forcing possibly nonexistent causal structure into the model may be detrimental to the models predictive capabilities.

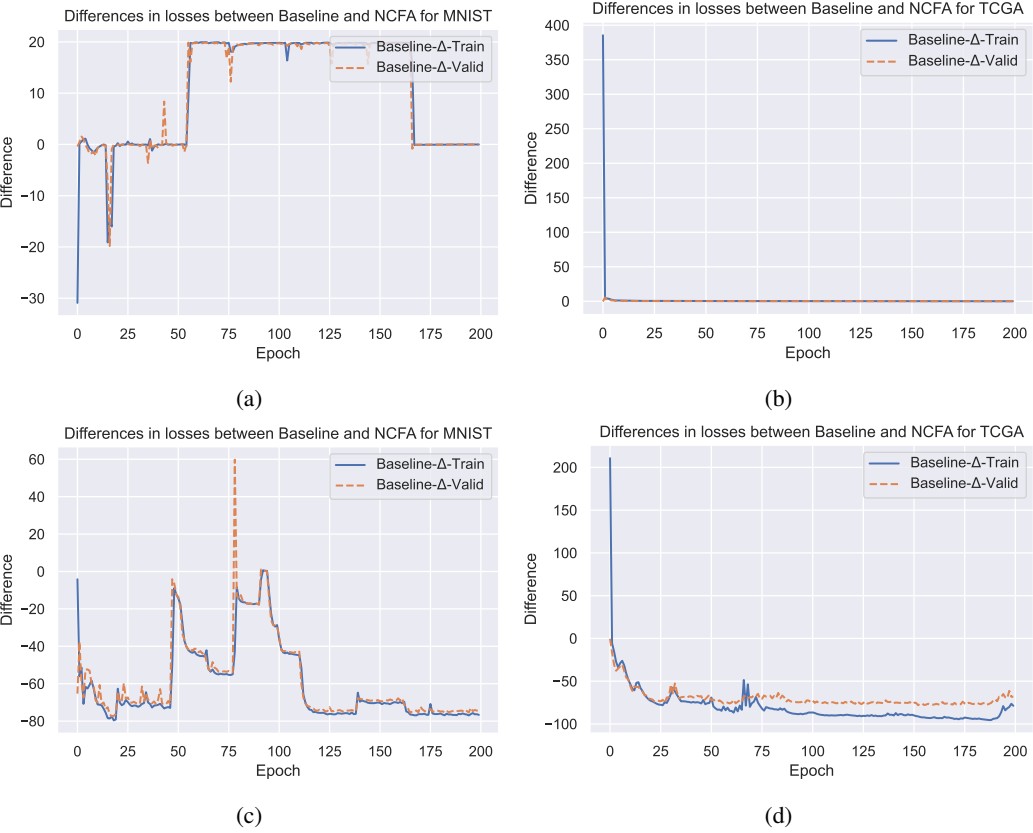

Figure 9: $\Delta$ curves for NCFA versus baseline VAE on MNIST and TCGA data sets. (a) and (b) show results of NCFA with default settings (on MNIST and TCGA respectively), while (c) and (d) show results of NCFA with additional 1-pure-child assumption (for both) and more sensitive independence tests (for MNIST).

---

[3]Instead of Chaterjee's coefficient based tests—it is known that distance covariance can be more powerful when one is concerned with finding independence, whereas Chatterjee's coefficient may be preferred for measuring strength of dependence (Chatterjee, 2022; Lin & Han, 2022).

