# OpenReview forum: "Neuro-Causal Factor Analysis"
_ICLR.cc/2024/Conference — Submitted to ICLR 2024_

### Official Review · Reviewer_VCXV · 2023-10-28

**Soundness:** 3 good
**Presentation:** 3 good
**Contribution:** 2 fair
**Rating:** 6
**Confidence:** 2

**Summary:**

The authors proposed Neuro-Causal Factor Analysis (NCFA) method, which is a nonparametric approach for modeling latent causal factors in deep generative models. They have proved theoretical results for the proposed method. They proposed an algorithm and applied it to simulated and real datasets. They compared their method with standard VAEs and showed that their method performs comparably and is more causally interpretable.

**Strengths:**

The paper is clearly written and theoretically solid.

**Weaknesses:**

The experiment section is a bit weak, and it lacks comparisons with other causal inference methods for deep generative models.

**Questions:**

- how sensitive the results are to the hyperparameters? and in practice, how should users choose them?

- is NCFA scalable for larger datasets and denser casual connections? what's the running speed of this method?

- how does NCFA compare to other causal inference methods for deep generative models?

---

> ### Author Response · Authors · 2023-11-21
>
> > how sensitive the results are to the hyperparameters? and in practice, how should users choose them?
>
> The main hyperparameter, lambda, is similar to the choice of latent dimension in any generative model: If lambda is too small, then we would run into the same kinds of problems training any generative model with insufficient capacity. It does not have to be tuned carefully in practice.
>
> More precisely, lambda should be greater than the number of causal latents, but how much greater is not as important. In practice (and our experiments on real data), VAEs are likely to have high enough latent DoF that experimentally exploring this further is beyond the scope of our presentation of the NCFA framework. Also see our response to the third question of Reviewer WRCe for discussion of how this parameter should be chosen by users.
>
> > is NCFA scalable for larger datasets and denser casual connections? what's the running speed of this method?
>
> Our real data application in Section 5 (especially Table 1, and further discussion in appendices F and C) demonstrates that NCFA scales to large data sets like MNIST, even with dense (fully connected) structures and up to a 1/4 of a million latent degrees of freedom in the VAE. The computational bottleneck is the NP-hard problem of finding an exact minimum edge clique cover, however our applications in Section 5 demonstrate that using polynomial-time ECC heuristics (based on the 1-pure-child assumption) bring the runtime to the order of standard VAE methods without sacrificing much of the accuracy of the exact method.
>
> > how does NCFA compare to other causal inference methods for deep generative models?
>
> This is discussed in Section 2.3. Our results are complementary to existing results, and make different, sometimes weaker, assumptions. We are happy to clarify any more specific questions you have on this.

---

> > ### Comment · Reviewer_VCXV · 2023-11-23
> > **Responses to authors' rebuttal**
> >
> > Thank the authors for their responses. I keep my current score.

---

### Official Review · Reviewer_eQDF · 2023-10-29

**Soundness:** 3 good
**Presentation:** 2 fair
**Contribution:** 2 fair
**Rating:** 3
**Confidence:** 3

**Summary:**

Neural Causal Factor Analysis (NCFA) was proposed in this submission to understand source-measurement causal relations in a factor analysis framework. By utilizing clique cover based latent structure constraints and variational autoencoder source-measurement mapping, the authors claimed that NCFA has the identifiability guarantee and also achieves nonlinear factor analysis with interpretable and flexible causal understanding. Experiments using simulated and real data (MNIST and TCGA) shows effective reconstruction of NCFA under certain settings.

**Strengths:**

NCFA emphasizes the identifiability guarantee with the focus on source-measurement two-level dependence structures.

With identified unconditional dependence graph (UDG), the authors reasoned that the corresponding minimum MCM graphs with minimum edge clique cover are identifiable under certain conditions, for example, 1-pure-child assumption.

NCFA leverages existing causal structure discovery and deep generative models to achieve computationally feasible source-measurement factor analysis.

The experiments showed that incorporating causal constraints, VAE has reasonable reconstruction performance.

**Weaknesses:**

The positioning of NCFA for causal structure discovery may be problematic as both causal understanding and interpretability are quite constrained.

It is not really clear how the identifiability guarantee may be beneficial for causal understanding under the constrained settings of NCFA.

The presented experimental results appear to be preliminary, without much explanations on causal relationships, interpretability, benefits from identifiability and flexibility by having nonlinear mapping using VAE.

The first part of NCFA depends on the quality of UDG and minimum MCM graphs. UDG appears to be a Markov Network for statistical independence representation. By the nature of Markov Networks, UDG can be guaranteed to be an I-map but not be a perfect map for the data generation distribution. The UDG and induced minimum MCM graphs therefore may not always capture the underlying causal structure faithfully. For example, when we have a M_1 to M_5 forming a Markov chain. Based on the described algorithm, the UDG may just have a complete graph with 5-node clique, which does not really provide any additional causal understanding or constraints in NCFA.

The authors may want to clearly define what they meant by NCFA being 'non-parametric' and 'identifiable'. For example, the overall FA framework is a parametric setting and VAEs also typically are considered parametric for the source-measurement mapping. If the authors meant that there is no need to specify mapping functional families as being 'non-parametric', it may need to explained clearly. It may also not be fair to claim the advantages on identifiability of minimum MCM graphs in NCFA over the existing other causal structure discovery methods due to the assumed specific settings.

Math notations may need careful proofread. For example, in Section 4, $\hat{U}$ was referred as 'estimated UDG' in the second and third paragraphs but then as the MCM graph at the beginning of the fourth graph. It is quite confusing. There are in fact many of these examples in the submission.

**Questions:**

1. What does MCM stand for? Medil Causal Models? If so, please provide the full name in the main text too.

2. What are the details of model structure and training, especially the VAE model constructed following the induced minimum MCM graphs? These do not appear to be included either in the main text or appendix.

---

> ### Author Response · Authors · 2023-11-21
>
> > What does MCM stand for? Medil Causal Models? If so, please provide the full name in the main text too.
>
> Yes, MCM stands for MeDIL causal model. We have added this to the main text.
>
> > What are the details of model structure and training, especially the VAE model constructed following the induced minimum MCM graphs? These do not appear to be included either in the main text or appendix.
>
> These details are given in Appendix C.
>
> > For example, when we have a M_1 to M_5 forming a Markov chain. Based on the described algorithm, the UDG may just have a complete graph with 5-node clique, which does not really provide any additional causal understanding or constraints in NCFA.
>
> Such graphs are not within the model class we are concerned with. Namely, your example is Markov to a DAG over the measurement variables, while we are explicitly concerned with causal factor models, which we motivate beginning in the second paragraph of Section 1.
>
> > The authors may want to clearly define what they meant by NCFA being 'non-parametric' and 'identifiable'.
>
> Our _statistical model_ is nonparametric (infinite-dimensional without functional restrictions), whereas the _estimator_ is parametric (finite-dimensional). The latter is of course always necessary if it is to be implemented on a computer, e.g. as with a neural network.  A similar phenomenon arises in density estimation: The model (usually H"older, Sobolev, etc.) is infinite-dimensional, but the estimator is finite-dimensional (e.g. kernel estimates). In other words, VAEs are parametric estimators of nonparametric models.
>
> >  It may also not be fair to claim the advantages on identifiability of minimum MCM graphs in NCFA over the existing other causal structure discovery methods due to the assumed specific settings.
>
> To the best of our knowledge, our identifiability results are the first of their kind for nonparametric causal factor models. Other methods make parametric assumptions and use this added power to relax their model class at the expense of stronger assumptions. Our approach has the advantage of not needing the stronger parametric assumptions, but of course if one is willing to make such assumptions for a given application, other methods are more appropriate.

---

> ### Comment · Reviewer_eQDF · 2023-11-22
>
> Thanks for providing clarifications. However, I still have concerns on limited experimental results and the positioning of the paper on the claim of causal structure discovery. I will have to keep the current score.

---

### Official Review · Reviewer_XmbE · 2023-10-30

**Soundness:** 2 fair
**Presentation:** 4 excellent
**Contribution:** 3 good
**Rating:** 5
**Confidence:** 3

**Summary:**

The paper combines advancements in causal discovery and deep learning, to propose a new nonparametric framework called Neuro-Causal Factor Analysis (NCFA). It uses variational autoencoder (VAE) with Markov factorization constraints of the distribution with respect to the learned graph for learning causal factors in a neuro mode.

**Strengths:**

1. The presentation is clear.
2. The experiements concerning the validation delta of the learning process is enough.

**Weaknesses:**

1. Some aspects concerning the theoretical soundness should be improved.
2. The efficiency should be further analyzed.

**Questions:**

1. Fig 2: it seems that all graphs can be aligned as the hierarchical structure. Is this an assumption or based on some theoretical results?
2. Definition 3.5: why this is called "observationally equivalent"? How about calling “equivalent wrt d-separation”?
3. Algorithm 1: it is clear that the partition of variables into "two groups" are critical to the performance of the algorithm. Is there any analysis about the efficiency, and the error cascade if an inaccuracy partition appears?

---

> ### Author Response · Authors · 2023-11-21
>
> > Fig 2: it seems that all graphs can be aligned as the hierarchical structure. Is this an assumption or based on some theoretical results?
>
> It is a well-known fact (due to Pearl) that any latent variable model can be represented minimally in this way. So this is not an assumption, and follows from known theoretical results.
>
> > Definition 3.5: why this is called "observationally equivalent"? How about calling “equivalent wrt d-separation”?
>
> The phrase "observationally equivalent" is standard in the causal graphical models literature. For example, see Judea Pearl's seminal book "Causality", which uses the phrase in Theorem 1.2.8 (on page 19, in the printing from 2000).
>
> > Algorithm 1: it is clear that the partition of variables into "two groups" are critical to the performance of the algorithm. Is there any analysis about the efficiency, and the error cascade if an inaccuracy partition appears?
>
> The algorithm does not partition variables. Line 2 of the algorithm along with Definition 3.2 rigorously describes how the latent variables are learned from the measurement variables. We do provide analysis about efficiency and robustness in the face of error—we describe this at the end of the "Synthetic data" analysis in Section 5, with the help of Figure 3, as well as in more detail and with more plots in Appendix E.

---

> > ### Comment · Reviewer_XmbE · 2023-11-22
> > **Thank you**
> >
> > Thank you for the rebuttal. I still think that hierarchical structure is an assumption (for example, contains acyclicity, no confounders etc), and some expressions like "observationally equivalent" should be rephrased more mathematically rigorous, like being "equal" with respect to Markov condition (or state this in terms of conditional probability given observational set S), and I remain unchanged of my score.

---

### Official Review · Reviewer_hmd4 · 2023-10-31

**Soundness:** 2 fair
**Presentation:** 1 poor
**Contribution:** 2 fair
**Rating:** 5
**Confidence:** 4

**Summary:**

This work proposes to use the VAE for modeling the factor analysis and aim to identify the factor via latent causal discovery.

**Strengths:**

- This work Neuro-Causal factor analysis using a VAE framework.

**Weaknesses:**

- Clarity is one of the main issues of this work. Many terminology descriptions are missing or without explanation. For example, what is the full name of MCM graph and ECC model?
- Moreover, the contribution of this work is rather limited and it seems to a simple incremental of the work Markham & Grosse-Wentrup, 2020.
- The notations are confusing and the theoretical results in this paper are problematic. Theorem 3.6 states that the DAG G is identifiable with the conditions that $M_{i}\perp M_{j}\iff i-j\notin E^{\mathcal{U}}$, which is problematic. Since a DAG identifiable means that every direction for every edge in the DAG will be identified, which is impossible without any assumptions under the existence of a latent variable. In fact, this theorem relies on the theory of Markham & Grosse-Wentrup, 2020 which has several underlying assumptions and constrained which is missing in the statement of the theorem.

**Questions:**

See the weaknesses above.

---

> ### Author Response · Authors · 2023-11-21
>
> > Clarity is one of the main issues of this work. Many terminology descriptions are missing or without explanation. For example, what is the full name of MCM graph and ECC model?
>
> MCM abbreviates "MeDIL causal model", and ECC abbreviates "edge clique cover". The MCM graph is rigorously defined in Definition 3.2 and ECC-observational equivalence is rigorously defined in Definition 3.5.
>
> > The notations are confusing and the theoretical results in this paper are problematic. Theorem 3.6 states that the DAG G is identifiable with the conditions that, which is problematic. Since a DAG identifiable means that every direction for every edge in the DAG will be identified, which is impossible without any assumptions under the existence of a latent variable. In fact, this theorem relies on the theory of Markham & Grosse-Wentrup, 2020 which has several underlying assumptions and constrained which is missing in the statement of the theorem.
>
> The underlying assumptions and theory required for Theorem 3.6 are clearly stated in the first sentence of the theorem, in the phrase "the data-generating distribution is Markov to a minimum MCM graph". Being Markov to a minimum MCM graph means that each clique in the ECC of U implies the existence of a latent variable in G (and this is clearly stated in the proof, which explicitly references all necessary supporting theory). This latent variable (by the assumption that the MCM graph is minimum), necessarily has exactly the edges directed to the measurement variables in its corresponding clique. Hence, the clearly stated assumption renders the latent DAG (and every direction of every edge in it) identifiable.

---

### Official Review · Reviewer_WRCe · 2023-11-01

**Soundness:** 2 fair
**Presentation:** 3 good
**Contribution:** 3 good
**Rating:** 5
**Confidence:** 4

**Summary:**

This work considers the classic factor analysis model from a causal perspective. The authors introduce a new method known as Neuro-Causal Factor Analysis. Unlike traditional factor analysis, the authors integrate a causal discovery approach and a variational autoencoder tool to uncover the latent causal graph and its corresponding parameters. They show the minimum  ﻿MeDIL Causal Model is identifiable under some conditions.
Furthermore, they validate the effectiveness of the proposed method through experiments conducted on both synthetic and real-world datasets.

**Strengths:**

1. The paper addresses a non-trivial task and introduces a new method to tackle this challenge. In contrast to most traditional factor analysis (FA) methods, the approach proposed in this paper can handle nonlinear FA models.

2. The identification results of this paper are novel and significant for the causal discovery and generative models community.

3. The experimental results are presented in a logical way.

**Weaknesses:**

1. I have some confusion regarding the conclusion in Theorem 3.6. This conclusion states that if there exist UDGs with a unique minimum edge clique cover, then we can uniquely identify that graph G. I attempted to read the proof; however, the author references the conclusions of two other articles to establish this point. Without any specific constraints on the generating function, the validity of this conclusion requires further elucidation. I hope the author can provide an intuitive proof framework to demonstrate the correctness of this conclusion.

2. The algorithm's results depend on the latent degree of freedom lambda.

**Questions:**

1. How can we ensure the identifiability of loading functions (f) and residual measurement errors?

2. Could you provide an intuitive proof framework to demonstrate the correctness of Theorem 3.6? Are there any graphical conditions when UDGs have a unique minimum edge clique cover?

3. In practice, how do we set the latent degree of freedom lambda?

minor typos:

 Section 2.2 on Page 4: which relax the Causal Markov Assumption? It should be the Causal Sufficiency Assumption, right?

Figure 2: double l_2,2?

---

> ### Author Response · Authors · 2023-11-21
>
> > How can we ensure the identifiability of loading functions (f) and residual measurement errors?
>
> Our focus is on identifying the causal structure. We have left identifiability of the parameters to future work (and would note that there is already extensive work on this problem). Our approach is to emphasize causal interpretability and accurate generative modeling as opposed to factor loading identification.
>
> > Could you provide an intuitive proof framework to demonstrate the correctness of Theorem 3.6? Are there any graphical conditions when UDGs have a unique minimum edge clique cover?
>
> In Corollary 3.7 (and its proof in Appendix B), we prove that a UDG has a unique minimum ECC when it has equal intersection and independence numbers. We also show that this condition on UDGs is equivalent to the 1-pure-child condition on latent DAG models known in the literature.
>
> > In practice, how do we set the latent degree of freedom lambda?
>
> The experiments show that this parameter does _not_ need to be carefully tuned, and indeed this is a key contribution. Simply put, we need lambda to be larger than the true number of causal latents, and small enough to train in practice. In other words, lambda could be in the thousands and it does not matter very much. (If lambda is too small, then we would run into the same kinds of problems training any generative model with insufficient capacity.)
>
> > Section 2.2 on Page 4: which relax the Causal Markov Assumption? It should be the Causal Sufficiency Assumption, right?
>
> The causal Markov assumption (CMA) is basically (as mentioned in the second sentence of Section 2.2) Reichenbach's common cause principle plus the causal sufficiency assumption (CSA), so relaxing the CSA is one (but not the only) way of relaxing the CMA.
>
> >   * Figure 2: double l_2,2?
>
> Yes, thanks for pointing out the typo! It should be l_2,1 and l_2,2.

---

> > ### Comment · Reviewer_WRCe · 2023-12-05
> > **Response**
> >
> > Thanks for your efforts. Having carefully reviewed the response and the comments from other reviewers, I will maintain my current score.

---

### Meta-Review · Area_Chair_Jnap · 2023-12-15

**Metareview:**

The paper considers a set of variables $M_i$ and defines the graph $\cal U$ (unconditional dependence graph) connecting two observed variables iff they are not independent.
Considering a minimum edge clique cover of $\cal U$, i.e. a set of subset $C_k$ of observed variables such that all variables in this subset are dependent on each other,
one associates to each $C_k$  a bunch of latent variables $L_{k,1}, \ldots L_{k,m}$
and considers the VAE defined by connecting variables $L_{k_i}$ to observed variables in $C_k$ only.

A main stated contribution is: "it comes with identifiability guarantees".
The condition for this guarantee is that $\cal U$ induces a single clique.

The paper has merits (analysis, discussion, open source algo).

**Justification For Why Not Higher Score:**

Contribution is meager. The paper is better written than 444 but the advance is smaller.

**Justification For Why Not Lower Score:**

NA

---

### Decision · Program_Chairs · 2024-01-16

Reject